# The role of nodes in controlling and observing complex networks

Longlong Wu *

School of Electromechanical Engineering, Jiangxi Vocational and Technical College of Communication, Nanchang, China

* wlljxjtxy@163.com

## Abstract

Dynamic processes on complex networks are closely associated with a variety of real-world systems. The controllability and observability of these networks are critical topics in the field of network science. Motivated by recent advancements in the study of structural controllability and observability, we investigate the roles of nodes in controlling and observing complex networks. Specifically, we categorize individual nodes into one of four types: driver nodes, sensor nodes, dual-identity nodes, and ordinary nodes. We propose a general framework for identifying the category of each node, thereby facilitating the exploration of the structural characteristics of these node types. Our findings indicate that these four types of nodes are prevalent in the control and observation of real networks. Through the analysis of their structural characteristics, we observe that nodes involved in controllability and observability are more likely to be low-degree nodes. Furthermore, we show that the proportions of these node categories are largely governed by the degree distribution of the network. Additionally, we present a theoretical analytical method to derive the proportions of the four node types, based on the network's degree distribution.

**Citation:** Wu L (2025) The role of nodes in controlling and observing complex networks. PLoS One 20(6): e0325824. https://doi.org/10.1371/journal.pone.0325824

**Data availability statement:** All relevant data are within the manuscript and its Supporting information files.

## Introduction

How to control and observe a network is an important issue in the analysis and study of complex networks and is also a research hotspot in network science [1–6]. People further developed classical theory to solve the control and observation problem of complex networks. At present, the research on the controllability and observability of complex networks has aroused widespread attention [7–11].

According to classical control theory [12], a network is called controllable if there exists an appropriate input that allows the state variables to reach any desired final state from the initial state in a finite time. Correspondingly, a system is called observable if, for each possible evolution of the system's state, the current state can be estimated using only the output information. Due to the large scale and high complexity of complex networks, traditional theories of controllability and observability are challenging to apply directly to network systems. Structural controllability and observability address this issue by analyzing network systems from the perspective of their topological structure [13]. A network system is considered structurally controllable and observable if there exists a set of specific values for independent free parameters

**Funding:** This work was supported by the Science and Technology Research Project of Jiangxi Provincial Department of Education under Grant GJJ2405313. The funders had no role in study design, data collection and analysis, decision to publish, or preparation of the manuscript.

**Competing interests:** The authors have declared that no competing interests exist.

in the state matrix, input matrix, and output matrix that satisfy the conditions for controllability and observability. Liu et al. [14] further studied the structural controllability of complex networks. They transformed the network controllability problem into a maximum matching problem and gave the minimum number of driver nodes required to control the network. The introduction of structural controllability has aroused a wave of research on network control. Many people have studied and analyzed the controllability and observability of complex networks from different perspectives and achieved fruitful results [15–21].

In this paper, we propose a novel framework for characterizing the functional roles of nodes in the structural controllability and observability of complex networks. Specifically, we propose an innovative classification scheme in which nodes are categorized based on their distinct roles in network control and observation: (1) driver nodes, which exert control the network dynamics, (2) sensor nodes, responsible for network observability, (3) dual-identity nodes, simultaneously functioning as both driver and sensor nodes, and (4) ordinary nodes, which do not participate directly in control or observation. Leveraging graph-theoretic principles, we develop a maximum matching-based framework capable of systematically identifying all four node categories in arbitrary directed networks. A key advancement of our approach is that all node types can be determined through a single maximum matching operation, significantly enhancing computational efficiency and theoretical clarity.

Through a comprehensive analysis of various real and model networks, we have several fundamental findings. First, we demonstrate the ubiquity of all four node categories in different network topologies, with dual-identity nodes representing a previously unrecognized but crucial component. Second, we further investigate the structural characteristics of four node categories and find that nodes involved in controllability and observability tend to prefer low-degree nodes, driver nodes preferentially select divergent (high out-degree) nodes, and sensor nodes tend to select convergent (high in-degree) nodes. More significantly, we establish that the proportions of four node categories are fundamentally governed by the network's degree distribution. Building upon these empirical observations, we develop a novel theoretical framework that enables the analytical prediction of category proportions directly from a network's degree distribution. Together, the proposed classification scheme and analytical tools open up new avenues for research in network science and control theory.

## Controllability and observability of complex networks

Given a linear time-invariant system [12], the system dynamics are described by the following equations:

$$
\begin{aligned}
\dot{X} &= AX + BU, \\
\dot{Y} &= CX,
\end{aligned}
\tag{1}
$$

where $X = \{x_1, \dots, x_N\}$ is the state vector, $U = \{u_1, \dots, u_p\}$ is the input vector, $Y = \{y_1, \dots, y_q\}$ is the output vector, $A = (a_{ij})_{N \times N}$ is the state matrix, $B = (b_{ij})_{N \times p}$ is the input matrix, and $C = (c_{ij})_{q \times N}$ is the output matrix.

Complex networks, represented as digraphs $G(A, B, C) = (V, E)$, can be modeled as linear time-invariant systems. The structure of the network is captured by two key components: the node set $V$ and the edge set $E$. The node set is partitioned into three disjoint subsets, $V = V_A \cup V_B \cup V_C$, each corresponding to a different type of node within the system. Specifically, the state node set, denoted $V_A = \{x_1, \dots, x_N\} := \{v_1, \dots, v_N\}$, comprises the state variables that govern the evolution of the system. The input node set, $V_B = \{u_1, \dots, u_p\} := \{v_{N+1}, \dots, v_{N+p}\}$, represents external sources of influence or control. The output node set, $V_C = \{y_1, \dots, y_q\} := \{v_{N+p+1}, \dots, v_{N+p+q}\}$, consists of the nodes that capture the observable outputs of the system.

The edge sets are defined as follows: $E_A = \{(x_j, x_i) \mid a_{ij} \neq 0\}$ is the set of edges between state nodes, $E_B = \{(u_j, x_i) \mid b_{ij} \neq 0\}$ is the set of edges from input nodes to state nodes, and $E_C = \{(x_j, y_i) \mid c_{ij} \neq 0\}$ is the set of edges from state nodes to output nodes.

In the study of structural controllability and observability, the matrices $A$, $B$, and $C$ in the system (1) are assumed to be structural matrices, containing only zero elements and independent free parameters. The system is said to be structurally controllable if it is possible to assign values to the independent free parameters such that the resulting system is controllable. This condition is satisfied if and only if the controllability matrix [13,14]

$$Q_c = [B, AB, A^2 B, \dots, A^{N-1} B],\tag{2}$$

has a generic rank equal to $N$, i.e., $\text{rank}_g(Q_c) = N$. Here, the generic rank of the structural matrix refers to the maximum rank that the matrix can achieve as a function of its independent free parameters. Similarly, the system is said to be structurally observable if the generic rank of the observability matrix [15]

$$Q_o = \left[ C^T, (CA)^T, (CA^2)^T, \dots, (CA^{M-1})^T \right]^T,\tag{3}$$

satisfies $\text{rank}_g(Q_o) = N$.

## The role of nodes

We introduce the relevant definitions of structural controllability and observability of complex networks.

**Definition 1.** *[14] In the structural controllability of complex network $G(A, B, C)$, state nodes in $V_A$ connected by external inputs are referred to as controlled nodes, and controlled nodes that do not share the same inputs are driver nodes. The number of controlled nodes is equal to the number of non-zero elements in the input matrix B, and the number of driver nodes corresponds to the dimension p of the input vector, i.e., the number of columns of the input matrix B.*

**Definition 2.** *[15] In the structural observability of complex network $G(A, B, C)$, the state nodes connected to outputs and do not share the same outputs are called sensor nodes. The number of sensor nodes corresponds to the dimension q of the output vector, i.e., the number of rows of the output matrix C.*

We investigate the role of individual nodes in controlling and observing the network by classifying each node into one of the following four categories: driver node, sensor node, dual-identity node, and ordinary node, where the dual-identity node is a node that functions as both a driver node and a sensor node, and is referred to as a DS node. For example, as shown in Fig 1, nodes $x_1$ and $x_2$ are driver nodes; nodes $x_2$ and $x_3$ are sensor nodes; and node $x_2$ is the DS node.

In order to give the method for identifying the role of nodes, we introduce the concepts of matching and maximum matching [22].

**Definition 3.** *For digraphs, a subset of edges $\mathcal{M}$ is called a matching if no two edges in $\mathcal{M}$ share a common start node or a common end node. The edges in $\mathcal{M}$ are referred to as matching edges. A node is considered matched if it is the end node of a matching edge; otherwise, the node is unmatched. For undigraphs, a subset of edges $\mathcal{M}$ is a matching if no two edges in $\mathcal{M}$ share a common node. The edges in $\mathcal{M}$ are referred to as matching edges. A node is considered matched if it is incident to a matching edge; otherwise, the node is unmatched.*

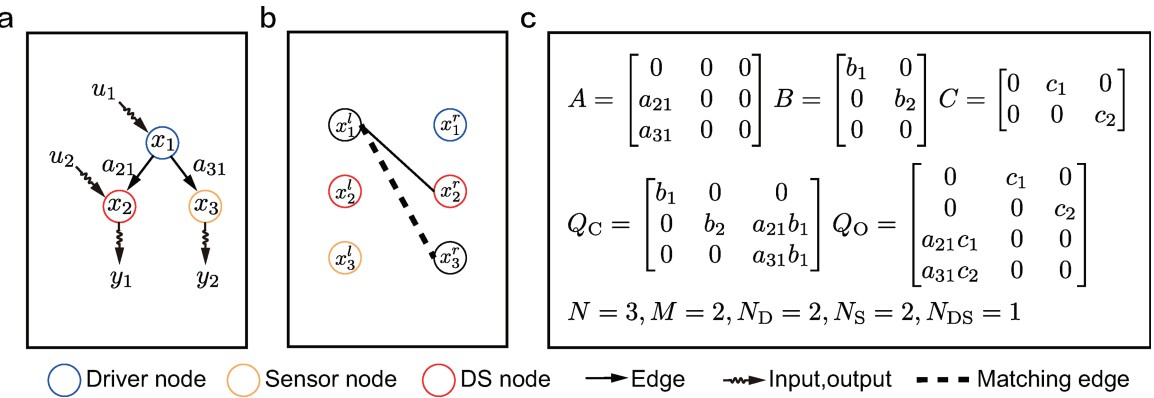

**Fig 1. Controllability and observability of complex networks.** (a) Controlling and observing a digraph $G(A,B,C)$. (b) The bipartite representation $H(A)$ and its maximum matching. (c) The controllability matrix and observability matrix.

Then we introduce the bipartite representation of a digraph $G(A)$ without inputs and outputs.

**Definition 4.** *[14] The bipartite representation of $G(A)$ is denoted as $H(A) = (V_A^l \cup V_A^r, E_A)$. The left node set $V_A^l = \{x_1^l, \ldots, x_N^l\}$ and the right node set $V_A^r = \{x_1^r, \ldots, x_N^r\}$ correspond to the $N$ columns and rows of the matrix $A$, respectively. The undirected edge set is defined as $E_A = \{(x_j^l, x_i^r) \mid a_{ij} \neq 0\}$.*

**Remark 1.** *The directed matching edges in G(A) correspond to the undirected matching edges in H(A), and the unmatched nodes in G(A) correspond to the unmatched nodes in the right node set $V_A^r$ of H(A).*

The maximum matching can be computed using the Hopcroft-Karp algorithm [23] based on MATLAB's Boost Graph Library [24], which has a time complexity of $O(\sqrt{|V|}|E|)$, where $|V|$ and $|E|$ denote the number of nodes and edges, respectively. This algorithm is highly efficient for sparse or moderately dense digraphs, but its performance degrades as network size and edge density increase.

We introduce the minimum input theory and the minimum output theory.

**Lemma 1.** *[14] For a complex network represented by the digraph G(A), the minimum number of driver nodes necessary to guarantee structural controllability is one, provided that there exists a perfect matching in G(A). When this condition holds, any single node can be chosen as a driver node. However, if a perfect matching is not present, the minimum number of driver nodes corresponds to the count of unmatched nodes in G(A). In such cases, the driver nodes are exactly those unmatched nodes.*

**Lemma 2.** *[15] For a complex network represented by the digraph G(A), the minimum number of sensor nodes required for structural observability is the same as the minimum number of driver nodes in its dual system. The dual system is derived by reversing the direction of all edges in G(A).*

Based on the above definitions and lemmas, we give the method for identifying the role of nodes.

**Theorem 1.** *For a complex network represented by a digraph G(A), its nodes are classified as driver node, sensor node, DS node and ordinary node when controlling and observing G(A).*

*The role of a node $v_i$ can be determined by the maximum matching of the binary representation H(A) of G(A):*

*1) A node $v_i$ is the driver node if and only if its corresponding node $v_i^r$ is the unmatched node in H(A);*

*2) A node $v_i$ is the sensor node if and only if its corresponding node $v_i^l$ is the unmatched node in H(A);*

*3) A node $v_i$ is the DS node if and only if its corresponding nodes $v_i^l$ and $v_i^r$ are both unmatched nodes in H(A);*

*4) A node $v_i$ is the ordinary node if and only if its corresponding nodes $v_i^l$ and $v_i^r$ are both matched nodes in H(A).*

**Proof 1**    *According to Lemma (1), the minimum number of driver nodes required to control a network G(A) is exactly the unmatched nodes in G(A). According to Definition (4) and Remark (1), the unmatched nodes in G(A) correspond to the unmatched nodes in the right node set $V_A^r$ of H(A). Therefore, a node $v_i$ is the driver node if and only if its corresponding node $v_i^r$ is the unmatched node in H(A). According to Lemma (2), the minimum number of sensor nodes required for structural observability is the same as the minimum number of driver nodes in its dual system, where the dual system is derived by reversing the direction of all edges in G(A), denoted as $G(A^T)$. According to Definition (4), bipartite representation $H(A^T)$ of the dual system of $G(A^T)$ is the same as H(A), except that the original $V_A^l$ and $V_A^r$ become $V_A^r$ and $V_A^l$, respectively. The unmatched nodes in $G(A^T)$ correspond to the unmatched nodes in the left node set $V_A^l$ of H(A). Therefore, a node $v_i$ is the sensor node if and only if its corresponding node $v_i^l$ is the unmatched node in H(A). The DS node is both the driver node and the sensor node, so a node $v_i$ is the DS node if and only if its corresponding nodes $v_i^l$ and $v_i^r$ are both unmatched nodes in H(A). Then the remaining nodes are ordinary nodes.*

For example, controlling and observing digraph $G(A)$ is shown in Fig 1(a). The bipartite representation $H(A)$ of $G(A)$ without input and output is shown in Fig 1(b). In Fig 1(c), the controllability and observability matrices are both full generic rank, indicating that the system is structurally controllable and observable. According to the maximum matching of the bipartite representation $H(A)$, $x_2^l$ and $x_3^l$ in the left node set $V_A^l$ are unmatched nodes, and $x_1^r$ and $x_2^r$ in the right node set $V_A^r$ are unmatched nodes. According to Theory (1), nodes $x_1$ and $x_2$ are driver nodes, nodes $x_2$ and $x_3$ are sensor nodes, and node $x_2$ is the DS node.

## Result

### The role of nodes in real networks

To demonstrate that the four types of nodes are ubiquitous in controlling and observing networks, we simulated 18 real-world networks, including regulatory networks, trust networks, food webs, electronic circuits, neural networks, citation networks, Internet, language networks, and transportation networks. Detailed information and sources for these networks are provided in Table 1.

We found that the four types of nodes are ubiquitous in controlling and observing real networks, as shown in Table 1. This indicates that, in the context of controlling and observing real networks, there are always nodes that assume dual roles. Moreover, we observed that the proportion of driver nodes $n_D$ is always equal to the proportion of sensor nodes $n_S$. Based on Lemmas (1) and (2), this symmetry is clearly evident in controlling and observing real networks. Furthermore, according to Theorem (1), we can deduce that driver nodes and sensor

**Table 1. Real network. For each network, we present its type, name, number of nodes *N*, number of edges *M*, and the fractions of driver nodes $n_D$, sensor nodes $n_S$, dual-identity nodes $n_{DS}$, and ordinary nodes $n_O$.**

| Type | No. | Name | N | M | $n_D$ | $n_S$ | $n_{DS}$ | $n_O$ |
|---|---|---|---|---|---|---|---|---|
| Regulatory | 1 | TRN-EC-2 [29] | 423 | 578 | 0.728 | 0.728 | 0.536 | 0.080 |
| | 2 | TRN-Yeast-1 [30] | 4684 | 15451 | 0.940 | 0.940 | 0.884 | 0.004 |
| | 3 | TRN-Yeast-2 [29] | 688 | 1079 | 0.821 | 0.821 | 0.658 | 0.016 |
| Trust | 4 | WikiVote [31] | 7115 | 103689 | 0.665 | 0.665 | 0.407 | 0.076 |
| Food Web | 5 | Grassland [32] | 88 | 137 | 0.522 | 0.522 | 0.250 | 0.205 |
| | 6 | Ythan [32] | 135 | 601 | 0.5111 | 0.511 | 0.185 | 0.163 |
| | 7 | Silwood [33] | 154 | 370 | 0.753 | 0.753 | 0.552 | 0.046 |
| Electronic circuits | 8 | s838a [29] | 512 | 819 | 0.232 | 0.232 | 0.045 | 0.580 |
| Neuronal | 9 | C. elegans [34] | 297 | 2359 | 0.164 | 0.165 | 0.057 | 0.727 |
| Citation | 10 | Small World [35] | 233 | 1988 | 0.601 | 0.601 | 0.343 | 0.142 |
| | 11 | SciMet [35] | 2729 | 10416 | 0.423 | 0.423 | 0.132 | 0.285 |
| | 12 | Kohonen [36] | 3772 | 12731 | 0.560 | 0.560 | 0.270 | 0.150 |
| Internet | 13 | Political blogs [37] | 1224 | 19090 | 0.356 | 0.356 | 0.157 | 0.445 |
| | 14 | p2p-1 [38] | 10876 | 39994 | 0.552 | 0.552 | 0.297 | 0.193 |
| | 15 | p2p-2 [38] | 8846 | 31839 | 0.578 | 0.578 | 0.322 | 0.1663 |
| Language | 16 | English words [39] | 7381 | 46281 | 0.634 | 0.634 | 0.423 | 0.155 |
| | 17 | French words [39] | 8325 | 24295 | 0.673 | 0.673 | 0.459 | 0.112 |
| Transportation | 18 | USair97 [40] | 332 | 2126 | 0.334 | 0.334 | 0.136 | 0.467 |

nodes correspond to the unmatched nodes in the left and right node sets of the bipartite representation $H(A)$ from the digraph $G(A)$. This symmetry is the fundamental reason why the proportion of driver nodes $n_D$ is equal to the proportion of sensor nodes $n_S$.

A noteworthy phenomenon is that the sum of the proportions of the four types of nodes in Table 1 exceeds 1. This is due to the fact that some nodes are classified into multiple categories. For instance, a DS node is both a driver node and a sensor node, resulting in it being counted twice. Consequently, we have the relationship $n_D + n_S - n_{DS} + n_O = 1$ in Table 1.

In Fig 2, we present the proportions of unique identities for the four types of nodes, where $n_D$ and $n_S$ represent the proportions of nodes that are exclusively driver nodes or sensor nodes, respectively. Clearly, $n_{DS}$ represents the overlapping portion of the driver node and sensor node proportions, and the sum of the four proportions equals 1. Upon comparison, we observe that the proportion of DS nodes in most real networks is relatively large, often exceeding 50%. This highlights the significant role of dual-identity nodes in controlling and observing real networks.

## Node characteristic

We primarily focus on the structural characteristics of four distinct types of nodes. The degree of a node in a network refers to the number of edges incident to that node. A higher degree signifies a greater number of neighboring nodes, indicating the node's increased significance within the network. We investigate whether the four types of nodes exhibit a tendency to preferentially select high-degree nodes, that is, nodes with a large number of connections.

We first introduce two types of model networks: the Erdős-Rényi (ER) random network [25] and the scale-free (SF) network [26]. The process of constructing a directed ER network begins with *N* isolated nodes, each assigned an identical weight of $1/N$, ensuring a uniform selection probability across all nodes. In each step, two distinct nodes $v_i$ and $v_j$ are randomly selected, and a directed edge $e_{ij}$ is established from node $v_i$ to node $v_j$ if no edge already

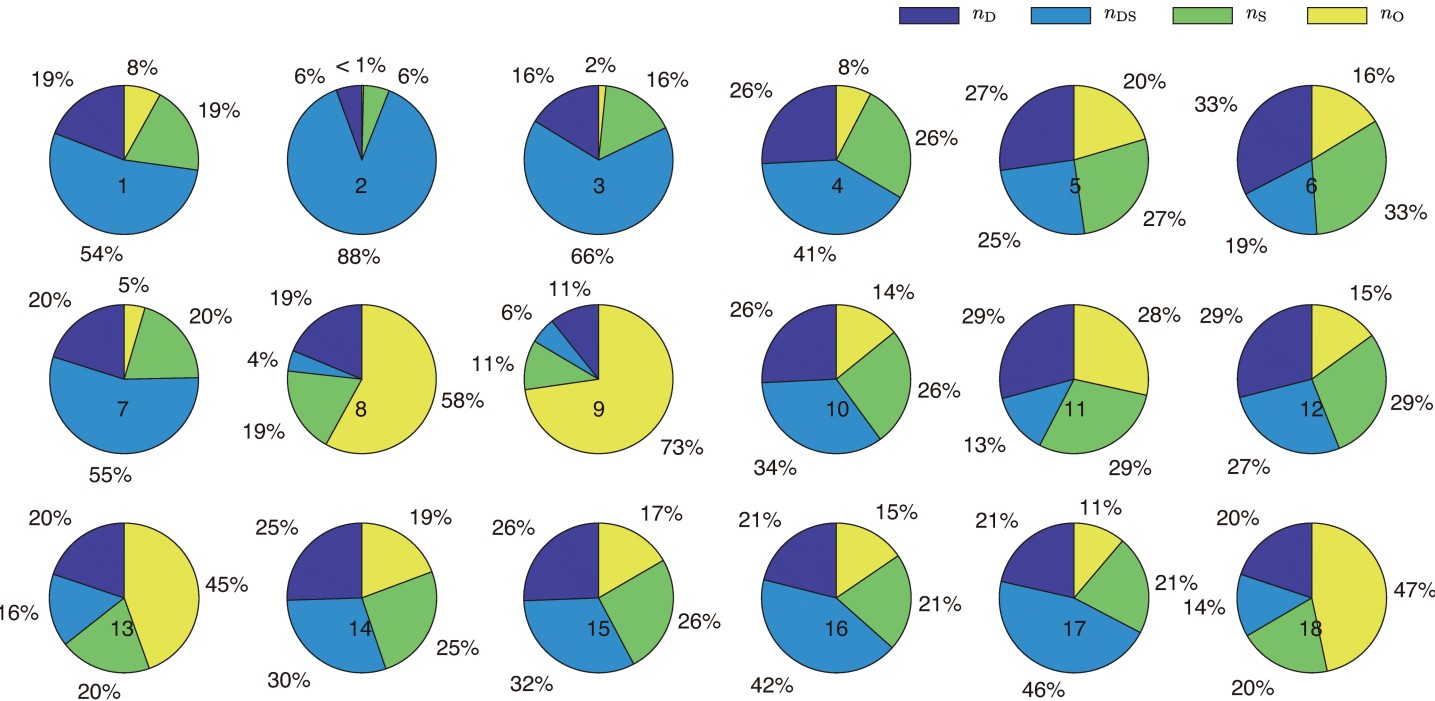

**Fig 2. Simulation results of real networks.** The proportion of unique identities of driver nodes $n_D$, sensor nodes $n_S$, dual-identity nodes $n_{DS}$, and ordinary nodes $n_O$, where $n_D$ and $n_S$ represent the proportions of nodes that are exclusively driver nodes or sensor nodes, respectively.

exists between them. This procedure is repeated until exactly $M$ directed edges are formed in the network, i.e., $|E| = M$.

Similarly, a directed SF network is constructed by starting with $N$ isolated nodes. Each node is assigned two weights, $p_i = i^{-a_{out}}$ and $q_i = i^{-a_{in}}$ ($i = 1, \dots, N$), where $a_{in}, a_{out} \in (0, 1)$. In each step, two nodes $v_i$ and $v_j$ are selected with probabilities $p_i/\sum_k p_k$ and $q_j/\sum_k q_k$ ($k = 1, \dots, N$), respectively. A directed edge $e_{ij}$ is then created from node $v_i$ to node $v_j$ if no edge already exists between them. This process continues until exactly $M$ directed edges have been established in the network, i.e., $|E| = M$.

We calculate the in-degree $k_v^-$ and out-degree $k_v^+$ of each node to obtain the average degree $\langle k \rangle = \langle k^- \rangle = \langle k^+ \rangle = M/N$ of the entire network, where $\langle k^- \rangle = \sum_{i=1}^{N} k_i^-$ and $\langle k^+ \rangle = \sum_{i=1}^{N} k_i^+$. Subsequently, we calculate the average degree for the different types of nodes: the driver nodes $\langle k_D^+ \rangle + \langle k_D^- \rangle$, sensor nodes $\langle k_S^+ \rangle + \langle k_S^- \rangle$, DS nodes $\langle k_{DS}^+ \rangle + \langle k_{DS}^- \rangle$, and ordinary nodes $\langle k_O^+ \rangle + \langle k_O^- \rangle$.

We compare the average degree of driver nodes, sensor nodes, DS nodes, and ordinary nodes in real networks with the overall average degree $\langle k^+ \rangle + \langle k^- \rangle$ of these networks. The simulation results are presented in Fig 3(a)–3(d). An anomalous observation is that the average degree of driver nodes, sensor nodes, and DS nodes is significantly smaller than $\langle k^+ \rangle + \langle k^- \rangle$ of the entire network. This suggests that the nodes involved in controlling and observing the network tend to preferentially select low-degree nodes. In contrast, the average degree of ordinary nodes $\langle k_O^+ \rangle + \langle k_O^- \rangle$ is significantly higher than that of the entire network, implying that high-degree nodes are generally not directly connected to external inputs and outputs. The simulation results based on the ER and SF model networks are shown in Fig 3(e)–3(h).

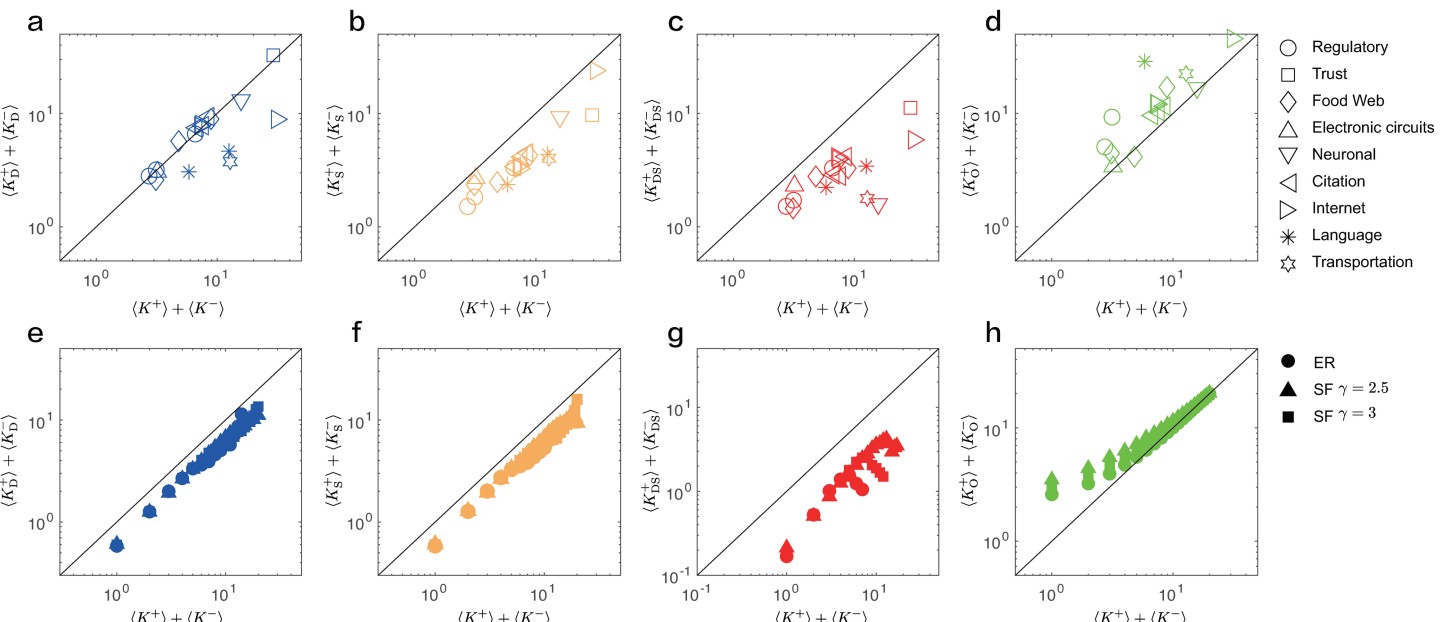

**Fig 3. Characteristics of four types of nodes.** (a-d) Comparison of the average degree of driver nodes $\langle k_D^+ \rangle + \langle k_D^- \rangle$, sensor nodes $\langle k_S^+ \rangle + \langle k_S^- \rangle$, DS nodes $\langle k_{DS}^+ \rangle + \langle k_{DS}^- \rangle$ and ordinary nodes $\langle k_O^+ \rangle + \langle k_O^- \rangle$ in real networks with the overall average degree $\langle k^+ \rangle + \langle k^- \rangle$ of real networks. (e-h) Comparison of the average degree of the four types of nodes in the ER and SF model networks with the overall average degree $\langle k^+ \rangle + \langle k^- \rangle$ of model networks.

We observe that the trends in these model networks are entirely consistent with those in real networks, further corroborating the finding that nodes involved in the control and observation of the network tend to preferentially select low-degree nodes.

We continue to examine the structural characteristics of the four distinct types of nodes. The in-degree $k_v^-$ and out-degree $k_v^+$ of nodes in the network are pervasive. Nodes can be classified into divergent nodes ($k_v^- < k_v^+$), convergent nodes ($k_v^- > k_v^+$), and balanced nodes ($k_v^- = k_v^+$). We investigate whether the four types of nodes exhibit a tendency to preferentially select divergent nodes, convergent nodes, or balanced nodes.

We compare the difference between the in-degree and out-degree for driver nodes $\langle k_D^+ \rangle - \langle k_D^- \rangle$, sensor nodes $\langle k_S^+ \rangle - \langle k_S^- \rangle$, DS nodes $\langle k_{DS}^+ \rangle - \langle k_{DS}^- \rangle$, and ordinary nodes $\langle k_O^+ \rangle - \langle k_O^- \rangle$ in real networks with the overall difference $\langle k^+ \rangle - \langle k^- \rangle = 0$ for real networks.

The simulation results based on real networks are presented in Fig 4(a)–4(d). We observe that driver nodes in almost all real networks satisfy $\langle k_D^+ \rangle - \langle k_D^- \rangle > 0$, indicating that driver nodes tend to preferentially select divergent nodes. In contrast, sensor nodes in most real networks exhibit $\langle k_S^+ \rangle - \langle k_S^- \rangle < 0$, suggesting that sensor nodes preferentially select convergent nodes. This further emphasizes the symmetry between driver and sensor nodes.

Furthermore, we find that the simulation results for DS nodes and ordinary nodes fluctuate around the value $\langle k^+ \rangle - \langle k^- \rangle = 0$, with the fluctuation for ordinary nodes being significantly smaller. This suggests that both divergent and convergent nodes are present in the DS node set, while the ordinary node set primarily consists of balanced nodes.

The simulation results based on the ER and SF model networks are presented in Fig 4(e)–4(h). We observe that the trends in these model networks closely mirror those in the real networks. For instance, the driver nodes in the model networks clearly exhibit $\langle k_D^+ \rangle - \langle k_D^- \rangle > 0$, and this trend becomes more pronounced as the average $\langle k \rangle$ increases. This further corroborates the finding that driver nodes tend to preferentially select divergent

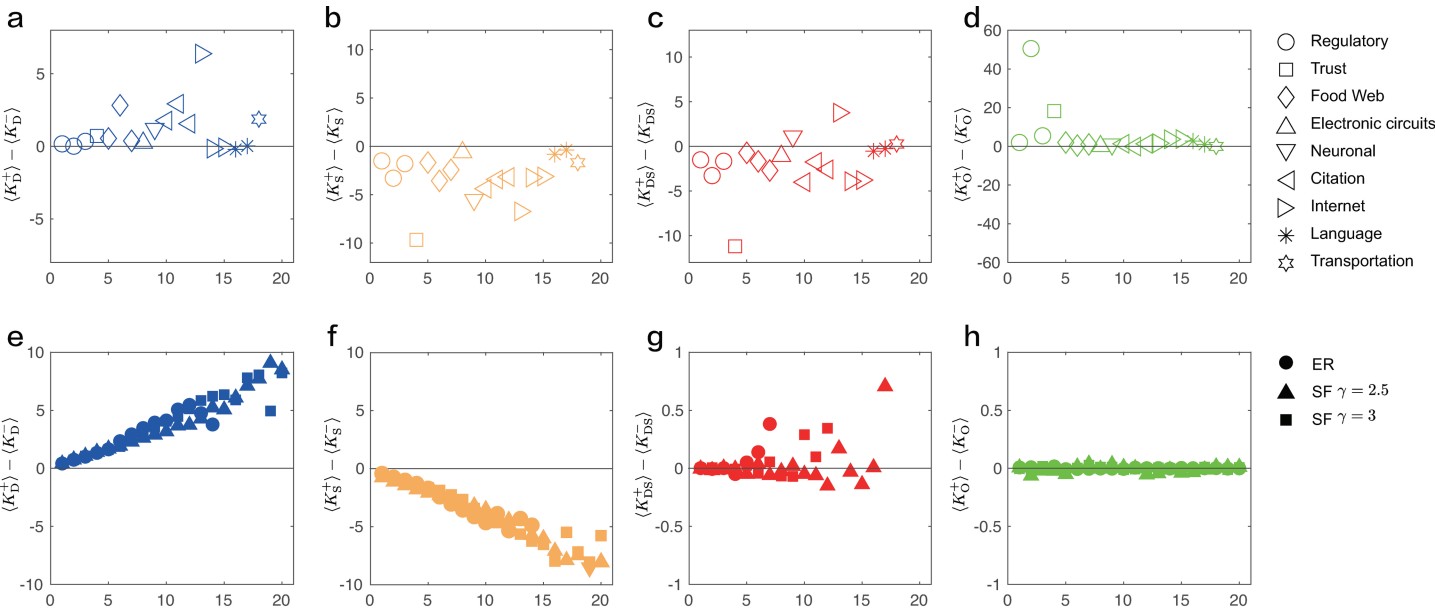

**Fig 4. Characteristics of four types of nodes.** (a–d) Comparison of the difference between the in-and-out degree of driver nodes $\langle k_D^+ \rangle - \langle k_D^- \rangle$, sensor nodes $\langle k_S^+ \rangle - \langle k_S^- \rangle$, DS nodes $\langle k_{DS}^+ \rangle - \langle k_{DS}^- \rangle$ and ordinary nodes $\langle k_O^+ \rangle - \langle k_O^- \rangle$ in real networks with the difference $\langle k^+ \rangle - \langle k^- \rangle = 0$ of real networks. (e–h) Comparison of the difference between the in-and-out degree of the four types of nodes in the ER and SF model networks with the difference of model networks.

nodes. Additionally, we observe that ordinary nodes in the model networks exhibit $\langle k_O^+ \rangle - \langle k_O^- \rangle \approx 0$, further confirming that ordinary nodes predominantly consist of balanced nodes.

## Analytical results of real networks

We investigate the network topology properties that determine the proportion of the four types of nodes and provide the analytical formulas for these proportions. To explore this, we employ the rand-degree network, which preserves the in-degree and out-degree of each node while randomly reconnecting the nodes. This procedure ensures that the degree distribution of the original network is retained, making the rand-degree network a useful reference in the simulations.

Fig 5(a)–5(c) illustrates the number of driver nodes $N_D$, DS nodes $N_{DS}$, and ordinary nodes $N_O$ in the real networks, compared with their corresponding proportions in the rand-degree networks. It is important to note that the number of driver nodes is exactly equal to the number of sensor nodes, i.e., $N_D = N_S$. Therefore, we present only the simulation results for the number of driver nodes in Fig 5. The simulation results demonstrate that the numbers of the four types of nodes in the real networks closely match those in the corresponding rand-degree networks. This suggests that the degree distribution of the network is a key attribute determining the number of the four types of nodes.

The dependence of $n_D$, $n_S$, $n_{DS}$, and $n_O$ on the degree distribution enables us to derive their analytical expressions. The theoretical framework is based on the core penetration theory [27], which is derived from the greedy leaf removal (GLR) procedure. In this procedure, each iteration removes: (1) a node with degree one (leaf) along with its sole nearest neighbor (root), and (2) the connecting edge, which is added to the matching. When multiple leaves connect to the same root, the algorithm arbitrarily selects one such pair for removal. For

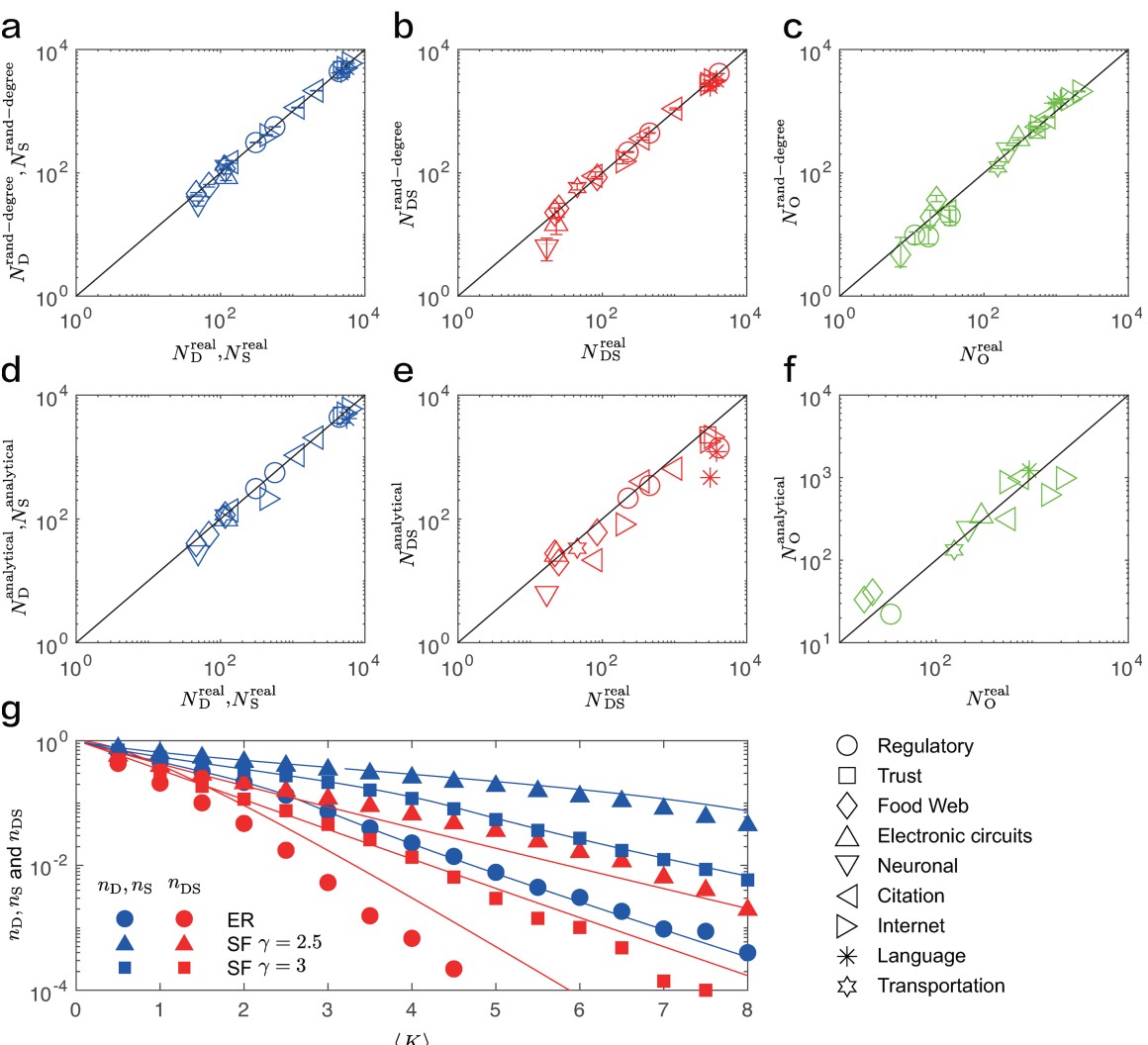

**Fig 5. Analytical results.** (a–c) Comparison of the number of driver nodes $N_D$, DS nodes $N_{DS}$ and ordinary nodes $N_O$ in the real networks with that of their corresponding rand-degree networks. (d–f) Comparison of the number of driver nodes $N_D$, DS nodes $N_{DS}$ and ordinary nodes $N_O$ in the real networks with that of analytical results. (g) The proportion of driver nodes $n_D$ and DS nodes $n_{DS}$ as the function of the average degree $\langle k \rangle$.

example, Fig 1(a)–1(b) give a digraph $G(A)$ and its bipartite representation $H(A)$, respectively. In $H(A)$, nodes $x_2^r$ and $x_3^r$ are leaves connected to root $x_1^l$. The GLR procedure would remove leaf $x_2^r$ and root $x_1^l$, subsequently adding edge $(x_1^l, x_2^r)$ to the matching. The GLR procedure can be implemented by referring to the LeetCode-solutions [28].

To establish our theoretical framework, we first present key graphical notions and probabilistic foundations. For an edge $(x_i^+, x_j^-)$ arriving at the node $x_i^+$ in the left node set $V_A^l$, let $\alpha^+$ and $\beta^+$ represent the probabilities that $x_i^+$ becomes a leaf or a root, respectively, in the GLR procedure. Similarly, for an edge $(x_i^+, x_j^-)$ arriving at the node $x_j^-$ in the right node set $V_A^r$, let $\alpha^-$ and $\beta^-$ represent the probabilities that $x_j^-$ becomes a leaf or a root, respectively, in the GLR procedure. For the networks with the degree distributions $P(k^-)$ and $P(k^+)$, we obtain the

following formula:

$$\alpha^{\pm} = \sum_{k^{\pm}=1}^{\infty} Q\left(k^{\pm}\right)\left(\beta^{\mp}\right)^{k^{\pm}-1}, \tag{4}$$

$$\beta^{\pm} = 1 - \sum_{k^{\pm}=1}^{\infty} Q\left(k^{\pm}\right)\left(1-\alpha^{\mp}\right)^{k^{\pm}-1}, \tag{5}$$

where $Q(k^{\pm}) = \frac{k^{\pm}P(k^{\pm})}{\langle k \rangle}$ and $\langle k \rangle = M/N$.

With the stable fixed solutions of $(\alpha^{\pm}, \beta^{\pm})$, we can calculate the proportion $n_c^+$ of nodes in the left node set $V_A^l$ and the proportion $n_c^-$ of nodes in the right node set $V_A^r$ that belong to the core, respectively.

$$n_c^+ = \sum_{k^+=0}^{\infty} P(k^+)\left[(1-\alpha^-)^{k^+} - (\beta^-)^{k^+}\right] - \langle k \rangle \alpha^+ (1-\alpha^- - \beta^-), \tag{6}$$

$$n_c^- = \sum_{k_-=0}^{\infty} P(k^-)\left[(1-\alpha^+)^{k^-} - (\beta^+)^{k^-}\right] - \langle k \rangle \alpha^- (1-\alpha^+ - \beta^+). \tag{7}$$

Additionally, we can determine the proportion $n_r$ of all roots in the left and right node sets.

$$n_r = \left[1 - \sum_{k^+=0}^{\infty} P\left(k^+\right)\left(1-\alpha^-\right)^{k^+}\right] + \left[1 - \sum_{k^-=0}^{\infty} P\left(k^-\right)\left(1-\alpha^+\right)^{k^-}\right] - \langle k \rangle \alpha^+ \alpha^-. \tag{8}$$

Since $n_r N$ represents the number of matching edges estimated from the roots of the GLR process, and $\min\{n_c^+, n_c^-\}N$ represents the number of matching edges estimated from the core structure of the GLR process, the maximum number of matching edges in the bipartite representation $H(A)$ is given by $n_r + \min\{n_c^+, n_c^-\}$. Thus, the proportion of driver nodes and sensor nodes is

$$n_D = n_S = 1 - \left(n_r + \min\{n_c^+, n_c^-\}\right). \tag{9}$$

Next, through extensive simulations of real and model networks, we derive an analytical estimate of the proportion of DS nodes by observing and summarizing the structural characteristics of DS nodes. Specifically, DS nodes are both driver nodes and sensor nodes, which makes them unmatched nodes in both the left and right node sets of $H(A)$. Let node $v$ with $k_v^- = 1$ and $k_v^+ = 0$ be classified as an in-leaf, and let the node with $k_v^- = 0$ and $k_v^+ = 1$ be classified as an out-leaf. Through numerous matching simulations, we observed that when multiple in-leafs share the same upstream neighbor node, these in-leafs have a high probability of being DS nodes. Similarly, when multiple out-leafs share the same downstream neighbor node, these out-leafs have a high probability of being DS nodes. Additionally, isolated nodes must be DS nodes. Based on the above observations and summaries, the proportion of DS nodes can be roughly estimated using the following formula

$$n_{DS} = P(0^-)P(0^+) + P(0^-)P(1^+) + P(1^-)P(0^+). \tag{10}$$

The analytical estimates of $N_D$, $N_{DS}$, and $N_O$ for real networks are derived from their degree distributions. For instance, we insert the in- and out-degree distributions of a real directed network into Eq (10) to predict $N_{DS}$ as follows:

$$N_{DS}^{\text{analytic}} = \left(P(0^-)P(0^+) + P(0^-)P(1^+) + P(1^-)P(0^+)\right)N, \tag{11}$$

where $P(0^{\pm})$ and $P(1^{\pm})$ represent the proportions of nodes in the real network with degree $k_v^{\pm} = 0$ and $k_v^{\pm} = 1$, respectively. The theoretical predictions for $N_D$ and $N_O$ are obtained in a similar manner.

Fig 5(d)–5(f) shows a comparison of the number of driver nodes $N_D$, DS nodes $N_{DS}$, and ordinary nodes $N_O$ in the real networks with their corresponding analytical results. The simulation results demonstrate that the numbers of the four types of nodes in the real networks closely match the analytical predictions. This validates the accuracy of our analytical estimates.

## Analytical results of model networks

We combine our analytical estimates with the degree distributions of the model networks to provide analytical results for the ER and SF networks. In ER networks, both the in- and out-degrees follow a Poisson distribution.

$$P(k) = P(k^-) = P(k^+) = e^{-\langle k \rangle} \frac{\langle k \rangle^k}{k!}. \tag{12}$$

By substituting the degree distribution of the ER network into the Formulas (4)–(9), we can obtain the analytical estimates of the proportion of driver nodes and sensor nodes.

The proportion of DS nodes is given as follows:

$$n_{DS}^{ER} = e^{-2\langle k \rangle}(2\langle k \rangle + 1). \tag{13}$$

Then the proportion of ordinary nodes is

$$n_{O}^{ER} = e^{-2\langle k \rangle}(2\langle k \rangle + 1) - 2[\alpha - \beta + \langle k \rangle \alpha(1-\beta)] + 1. \tag{14}$$

In SF networks, both the in- and out-degrees follow a power-law distribution given by

$$P(k) = P(k^-) = P(k^+) = \frac{[\langle k \rangle(1-a)]^{1/a}}{a} \frac{\Gamma(k-1/a, \langle k \rangle(1-a))}{\Gamma(k+1)}, \tag{15}$$

where $\Gamma(s, x)$ is the incomplete Gamma function, $\Gamma(n)$ is the Gamma function, and the power-law exponent $a = 1/(\gamma - 1)$. By substituting the degree distribution of the SF network into the Formulas (4)–(9), we can obtain the analytical estimates of the proportion of driver nodes and sensor nodes. Let $\delta$ denote $\frac{[\langle k \rangle(1-a)]^{1/a}}{a}$, and let $\Gamma_k$ denote $\frac{\Gamma(k-1/a, \langle k \rangle(1-a))}{\Gamma(k+1)}$. The proportion of DS nodes in SF networks is then given by:

$$n_{DS}^{SF} = \delta^2 \Gamma_0^2 + 2\delta^2 \Gamma_0 \Gamma_1. \tag{16}$$

Then the proportion of ordinary nodes is

$$n_{O}^{SF} = \delta^2 \Gamma_0^2 + 2\delta^2 \Gamma_0 \Gamma_1 - 2\langle k \rangle \alpha(1-\beta)(C_1 + C_2 + 1) + 3, \tag{17}$$

where the parameter $C_1 = \frac{\Gamma(-1/a, \langle k \rangle(1-a)\alpha)}{\Gamma(1-1/a, \langle k \rangle(1-a)\alpha)}$ and the parameter $C_2 = \frac{\Gamma(-1/a, \langle k \rangle(1-a)(1-\beta))}{\Gamma(1-1/a, \langle k \rangle(1-a)(1-\beta))}$.

Fig 5(g) shows the proportion of driver nodes $n_D$, sensor nodes $n_S$ and DS nodes $n_{DS}$ as the function of the average degree $\langle k \rangle$. We found that the proportion of the three types of nodes decreases rapidly with the increase of the average degree $\langle k \rangle$, and decreases with the increase

of the power exponent $\gamma$. This shows that dense (large $\langle k \rangle$) and homogeneous (large $\gamma$) networks are easier to control and observe. We also found that the analytical results are in good agreement with the simulation results, especially in more dense and homogeneous networks. This further illustrates the accuracy of our analytical estimates.

## Conclusion

We investigate the role of nodes in controlling and observing complex networks. Specifically, we classify individual nodes into four categories: driver nodes, sensor nodes, dual-identity nodes, and ordinary nodes. To identify the category of each node, we propose a general framework based on maximum matching, which facilitates the exploration of the structural characteristics of these four types. Our analysis demonstrates that these four types of nodes are prevalent in controlling and observing real networks. By studying the structural characteristics of these nodes, we find that those nodes involved in control and observation tend to preferentially select low-degree nodes.

A key finding of this study is that the proportion of the four types of nodes is strongly influenced by the degree distribution of the network. Additionally, we propose a theoretical analysis method to derive analytical results concerning the proportions of these four types of nodes.

## Supporting information

**S1 File. Real-world network datasets.** The 18 real-world network datasets used in this study. (ZIP)

## Author contributions

**Conceptualization:** Longlong Wu.

**Data curation:** Longlong Wu.

**Formal analysis:** Longlong Wu.

**Funding acquisition:** Longlong Wu.

**Investigation:** Longlong Wu.

**Methodology:** Longlong Wu.

**Project administration:** Longlong Wu.

**Resources:** Longlong Wu.

**Software:** Longlong Wu.

**Supervision:** Longlong Wu.

**Validation:** Longlong Wu.

**Visualization:** Longlong Wu.

**Writing – original draft:** Longlong Wu.

**Writing – review & editing:** Longlong Wu.

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
