## [Decision Letter · Decision Letter 0]

PONE-D-25-05744The Role of Nodes in Controlling and Observing Complex NetworksPLOS ONE

Dear Dr. Wu,

Thank you for submitting your manuscript to PLOS ONE. After careful consideration, we feel that it has merit but does not fully meet PLOS ONE’s publication criteria as it currently stands. Therefore, we invite you to submit a revised version of the manuscript that addresses the points raised during the review process.

We look forward to receiving your revised manuscript.

Kind regards,

Rossana Mastrandrea

Academic Editor

PLOS ONE

Journal Requirements:

3. Thank you for stating the following in the Acknowledgments Section of your manuscript: [This work was supported by the Science and Technology Research Project of Jiangxi Provincial Department of Education under Grant GJJ2405313.]

Please remove any funding-related text from the manuscript and let us know how you would like to update your Funding Statement. Currently, your Funding Statement reads as follows: “The authors received no specific funding for this work.”

Reviewers' comments:

Reviewer's Responses to Questions

**Comments to the Author**

1. Is the manuscript technically sound, and do the data support the conclusions?

Reviewer #1: Partly

Reviewer #2: Yes

2. Has the statistical analysis been performed appropriately and rigorously? 

Reviewer #1: Yes

Reviewer #2: Yes

3. Have the authors made all data underlying the findings in their manuscript fully available?

Reviewer #1: Yes

Reviewer #2: No

4. Is the manuscript presented in an intelligible fashion and written in standard English?

Reviewer #1: Yes

Reviewer #2: Yes

5. Review Comments to the Author

Reviewer #1: In this paper, authors have studied the role of nodes in controlling and observing

complex networks. This paper is written concisely and the topic sounds interesting. But

I still have some questions about the paper and my detailed comments are as follows:

1. The authors have categorized the nodes into four categories. Did you invent this

classification? What is the basis for such a categorization?

2. In Introduction, I can only see that this topic has a certain degree of researchability,

but it does not highlight the innovation of this paper.

3. The paper lacks relevant literature citations. For example, Equation (2) and Equation

(3) are derived without citation or proof.

4. The authors obtained theoretical results by using the maximum matching method,

what is the basis for using this method?

5. I noticed that the authors have simulated in all 18 real networks. Can the theoretical

results be visualized with real examples? Or do the theoretical results in this paper

have any physical meaning?

I think the paper should be revised.

Reviewer #2: Your manuscript provides a meaningful contribution to network controllability and observability by introducing a clear node classification framework and analytical methods. The findings are well-supported by simulations, and the symmetry between driver and sensor nodes is an interesting insight. Below are suggestions to improve the manuscript:

1. DS Node Identification. The heuristic for DS nodes (based on in-leaves/out-leaves) is plausible but lacks theoretical justification (equation 10).Provide a more rigorous derivation

2. Scalability. The maximum matching approach (Hopcroft-Karp) has known complexity limits for very large/dense networks. Discuss how the framework performs on networks with millions of nodes or high average degree.

3. Reproducibility. While the methodology is well-explained, sharing pseudocode or code for key steps (e.g., GLR procedure, bipartite matching) would aid reproducibility. Suggestion: Include an appendix or link to an open-source implementation.

4. Clarity: The writing is clear, but some technical passages (e.g., GLR procedure) could benefit from additional examples.

6. PLOS authors have the option to publish the peer review history of their article (what does this mean?). If published, this will include your full peer review and any attached files.

Reviewer #1: No

Reviewer #2: No

---

## [Author Response · Author response to Decision Letter 1]

Dear Reviewers:

I sincerely thank you for your valuable comments and suggestions to improve our paper. According to your comments, we have carefully revised the manuscript. In addition, I have answered your questions one by one. Hopefully, the revised manuscript is satisfactory now.

Best Wishes!

Author: Wu Longlong

---

## [Decision Letter · Decision Letter 1]

The Role of Nodes in Controlling and Observing Complex Networks

PONE-D-25-05744R1

Dear Dr. Wu,

We’re pleased to inform you that your manuscript has been judged scientifically suitable for publication and will be formally accepted for publication once it meets all outstanding technical requirements.

Kind regards,

Rossana Mastrandrea

Academic Editor

PLOS ONE

Additional Editor Comments (optional):

Reviewers' comments:

Reviewer's Responses to Questions

**Comments to the Author**

1. If the authors have adequately addressed your comments raised in a previous round of review and you feel that this manuscript is now acceptable for publication, you may indicate that here to bypass the “Comments to the Author” section, enter your conflict of interest statement in the “Confidential to Editor” section, and submit your "Accept" recommendation.

Reviewer #1: All comments have been addressed

Reviewer #2: All comments have been addressed

2. Is the manuscript technically sound, and do the data support the conclusions?

Reviewer #1: Yes

Reviewer #2: Yes

3. Has the statistical analysis been performed appropriately and rigorously? 

Reviewer #1: Yes

Reviewer #2: Yes

4. Have the authors made all data underlying the findings in their manuscript fully available?

Reviewer #1: Yes

Reviewer #2: Yes

5. Is the manuscript presented in an intelligible fashion and written in standard English?

Reviewer #1: Yes

Reviewer #2: Yes

6. Review Comments to the Author

Reviewer #1: (No Response)

Reviewer #2: (No Response)

7. PLOS authors have the option to publish the peer review history of their article (what does this mean?). If published, this will include your full peer review and any attached files.

Reviewer #1: No

Reviewer #2: No

---

## [Editor Report · Acceptance letter]

PONE-D-25-05744R1

PLOS ONE

Dear Dr. Wu,

I'm pleased to inform you that your manuscript has been deemed suitable for publication in PLOS ONE. Congratulations! Your manuscript is now being handed over to our production team.

Kind regards,

on behalf of

Dr. Rossana Mastrandrea

Academic Editor

PLOS ONE